# Automatic Dynamic Range Adjustment for Pedestrian Detection in Thermal (Infrared) Surveillance Videos

**DOI:** 10.3390/s22051728

**Published:** 2022-02-23

**Authors:** Oluwakorede Monica Oluyide, Jules-Raymond Tapamo, Tom Mmbasu Walingo

**Affiliations:** School of Engineering, University of KwaZulu-Natal, Durban 4041, South Africa; 213554623@stu.ukzn.ac.za (O.M.O.); walingo@ukzn.ac.za (T.M.W.)

**Keywords:** infrared surveillance, pedestrian detection, video surveillance

## Abstract

This paper presents a novel candidate generation algorithm for pedestrian detection in infrared surveillance videos. The proposed method uses a combination of histogram specification and iterative histogram partitioning to progressively adjust the dynamic range and efficiently suppress the background of each video frame. This pairing eliminates the general-purpose nature associated with histogram partitioning where chosen thresholds, although reasonable, are usually not suitable for specific purposes. Moreover, as the initial threshold value chosen by histogram partitioning is sensitive to the shape of the histogram, specifying a uniformly distributed histogram before initial partitioning provides a stable histogram shape. This ensures that pedestrians are present in the image at the convergence point of the algorithm. The performance of the method is tested using four publicly available thermal datasets. Experiments were performed with images from four publicly available databases. The results show the improvement of the proposed method over thresholding with minimum-cross entropy, the robustness across images acquired under different conditions, and the comparable results with other methods in the literature.

## 1. Introduction

Video surveillance is gaining worldwide prevalence, and it is not uncommon to find cameras mounted in airports, schools, office buildings, and many private residential areas in addition to their traditional presence on public and government buildings and large organisations. This significant increase in interest and utilization of Video Surveillance Systems (VSSs) outside of the public and security sectors is propelled by user demand for security due to increasing crime rates, global security threats, and advancement in technology, which has significantly dropped the cost of acquiring and ease of installing VSS. The consensus in demand is for the VSS to be proactive and persistent in monitoring. The most prevalent VSSs employ visible light cameras that do not easily function in a persistent—all day, all night, and every day of the week—manner and their performance is hampered by over, under, and uneven illumination during the day and the use of artificial light at night. Thermal infrared cameras can monitor persistently because they are not affected by the problems of visible light cameras. All objects, whether hot, at room temperature, or frozen, emit energy from the infrared part of the electromagnetic spectrum called heat signature. Sensors in thermal cameras can detect the objects’ infrared radiation and create an image based on that information.

Proactive VSS systems predict critical events before they happen and provide alerts for them. Pedestrian detection is a significant task towards achieving proactive systems as it provides primitive information for interpretation of the video footage. Pedestrians are typically warmer than most objects in thermal video streams because human skin emits (radiates) infrared energy almost perfectly with emissivity at 0.98 where 1 is the value of the perfect radiator [1]. The amount of infrared energy detected from objects in a scene by thermal sensors depends on the emissivity of the object, the reflectivity of other objects in the immediate vicinity [2], and the prevailing weather conditions. Figure 1 presents a scenario where the hand and the ring on the finger of the hand have the same temperature, but the ring appears cooler than the hand because the properties of the ring are different from the hand. In outdoor infrared surveillance footage, the appearance of the pedestrian is affected by the prevailing weather condition. This constitutes a major challenge for pedestrian detection.

Thermal surveillance cameras are normally set to capture as much of the scene data as possible through a process of tone-mapping based on the sensitivity of the Focal Plane Arrays (FPA) or thermal sensors to the magnitude of infrared energy present in a scene. Tone-mapping is, essentially, rendering high dynamic range data to low-dynamic range displays. Concerning thermal imaging, the dynamic range of a scene is the ratio of the hottest to the coldest region in the scene, the dynamic range of the camera is the ratio of the highest to lowest temperature the FPA or thermal sensors are capable of detecting and the dynamic range of the display is the ratio of the maximum to minimum intensities emitted from the screen. The dynamic range of the scene is far wider than the camera and the display; therefore, the scene is first tone-mapped to the dynamic range of the camera and then tone-mapped to the dynamic range of the display. As there is no specific target in mind, the resulting images have poor contrast and an unsuitable amount of detail for detecting a particular object. Before capturing a scene, the expected image can be improved, with better contrast and reduced details, by narrowing the dynamic range of the camera to be within the range of infrared emissions of the target in the scene so that a lower dynamic range is mapped to a wider dynamic range. If the dynamic range is not adjusted before scene capture, it is possible to manually adjust the image’s range to the range of the target and remap the image as shown in Figure 2. However, manual adjustment is not feasible for a video surveillance system handling large amounts of data.

The contribution of this work is a novel algorithm for candidate generation or ROI extraction, which adapts general-purpose techniques to the specific task of pedestrian detection in infrared images with a clear start and stopping criteria that efficiently suppresses the background to identify regions-of-interest for pedestrian detection in infrared images acquired under various weather conditions and in the presence of noise. The advantages of the proposed method are as follows. First, it is an automatic candidate generation method as opposed to manual delineation and cropping of pedestrians, which is tedious and time consuming. Second, it overcomes the limitations of background model-based candidate generation methods such as the computational cost of using the entire video and the inability to detect motionless pedestrians.

The rest of the paper is organised as follows. Section 2 presents related works. Section 3 presents the proposed algorithm. Section 4 provides the experimental results and discussion. Section 5 presents the conclusion and future work.

## 2. Related Works

Substantial effort has been made towards pedestrian detection because humans are the principal instigators of noteworthy events in surveillance videos. Pedestrian detection in infrared images generally consists of two steps: candidate generation and validation. Candidate generation involves detecting the region-of-interest (ROI) where pedestrians are likely to be found, while candidate validation involves discriminating between pedestrians and non-pedestrians within areas demarcated as ROI. Many algorithms have been proposed for visible light images [4,5,6,7,8,9,10,11,12] but, because images from visible light and thermal infrared imaging have different characteristics, adaptations are necessary for visible light algorithms to perform well on infrared images.

Rajkumar and Mouli [13] proposed a method for detecting pedestrians comprising a background subtraction model, high-boost filtering, and local adaptive thresholding. Jeon et al. [14] fused the result of background subtraction—pixel difference image—with edge pixel information for each video frame to detect pedestrians. Jeyabharathi and Dejay [15] proposed a detection method based on Reflectional Symmetrical Patterns (RSP). Successive frame differencing is used as an initial processing step, while RSP is used to create a subspace for background modelling. The pedestrians are finally obtained by thresholding. Ma [16] made use of the background difference method where the background is obtained by adding all video frames together and by taking the average value to extract the target. Background modelling usually uses the entire video to build a reliable model.Therefore, some methods initially perform background suppression rather than background subtraction. Rajkumar and Mouli [17] combined mean and Laplacian of Gaussian (LoG) filtering to suppress the background and enhance pedestrians, after which morphological processing and local thresholding are used to extract likely candidate regions.

Thresholding is also used to suppress the background due to the contrast between pedestrians and background. Wu et al. [18] proposed a new criterion for finding the optimal threshold by modelling the object and background as normal cloud classes and constructing a criterion that compares the hyper-entropies of both cloud class models. The cloud model used assumes that object and background distributions follow a normal distribution, and the hyper-entropy of each cloud model measures how much each class deviates from a normal distribution. Manda et al. [19] proposed the use of an analogous function, in this case, the raised cosine distribution function, to determine the threshold value for thermal image segmentation. Manda et al. [20] approximated the one-dimensional histogram of the image to the transient response of the first-order linear circuit to find the threshold value for image segmentation. Approximating the histogram or assuming a distribution might over-fit images from one database.

Some methods depend on the creation and extraction of features that are fed into a binary—object/non-object—classifier [21,22,23,24,25,26,27,28,29]. Zhao et al. [21] placed more emphasis on underlying temperature information in infrared images and trained a temperature net for pedestrian detection. Krišto et al. and Tumas et al. [25,26] used YOLOv3 to detect pedestrians from infrared images in varying weather conditions. Gao et al. [23] performed feature extraction using a transfer-learned visual geometry group (VGG-19) CNN. My et al. [28] made use of YOLOv3 object detector for pedestrian detection in IR images by applying a generative data augmentation strategy. Li et al. [29] proposed an improvement to YOLOv5 called YOLO-FIRI to include IR image features such that the network is forced to learn more discriminative features, and small infrared object detection accuracy is improved. images. While these methods achieve state-of-the-art results when used on RGB images, they do not perform similarly on infrared. Moreover, the performance of models across different datasets depends on similarity to the training data [25].

Minimum cross-entropy algorithms have been used in both visible and multispectral images [6,30,31,32,33]. In this work, we adapt it for candidate generation in thermal infrared pedestrian detection. First, rather than using the image in its original form, the image’s histogram is equalised to become uniformly distributed. This is necessary because entropy increases when information is uniformly distributed. Moreover, equalising the image provides a spatial cohesiveness to the infrared image necessary for the success of the proposed algorithm. Secondly, given the wide dynamic range of infrared images, the sensitivity of the image is iteratively reduced and the algorithm is “guided” towards the right side of the histogram. Therefore, at each iteration, the gray level at which minimum cross-entropy is attained becomes the new minimum dynamic range value for the image, thus creating a modified histogram for the next iteration until the previous and current minimum values become equal. Thirdly, several algorithms rely on normalizing the pixel intensity to provide a measure of the probability, but histogram equalisation achieves the same effect. Therefore, the known probabilities are the equalised pixel intensity values of the original image.

## 3. Proposed Method

The overview of the proposed method is presented in Figure 3 and consists of two stages: candidate generation and candidate validation. In the candidate generation step, median filtering is first applied to the input image to reduce the effect of noise. This is followed by the automatic dynamic range adjustment consisting of histogram specification whereby a uniformly distributed histogram is specified and iterative histogram partitioning and remapping whereby, at every chosen threshold, the range from zero to that point reset to black. This reset image is then iteratively repartitioned and reset until convergence. It is important to note that the beginning of the dynamic range adjustment algorithm is histogram specification because it has the effect of reducing the dynamic range and is what guarantees that when the iterative histogram partitioning converges, it always contains pedestrian candidates. In our experiments, histogram specification was carried out using histogram equalisation, and iterative histogram partitioning was carried out using minimum cross-entropy. After dynamic range adjustments, the resulting image is sent for candidate validation, which outputs the image with pedestrians detected.

### 3.1. Dynamic Range Adjustment

The amount of information captured from a scene, which determines the radiometric resolution of the image responsible for the actual information contained in an image, is dependent on the sensitivity of the infrared sensors in the thermal camera. The sensitivity of the infrared sensor describes its ability to detect slight differences in infrared energy, which are encoded using bits and are displayed in varying tones of gray. A high level of gray corresponds to high infrared energy emissions and vice versa. The images are, therefore, made up of numbers ranging from zero (0) to one minus the maximum number of gray levels available, and the maximum gray level is determined by the number of bits used to represent infrared energy detected. To highlight pedestrians, it is necessary to reduce the sensitivity of the camera so that regions with infrared emissions lower than the minimum detectable by the thermal sensor appear as black and regions, with infrared emissions higher than the maximum temperatures appearing as white. In this paper, we perform this sensitivity reduction using histogram specification and iterative histogram partitioning. The pseudo-code for the Dynamic Range Adjustment algorithm is shown in Algorithm 1.
**Algorithm 1:** Dynamic Range Adjustment.
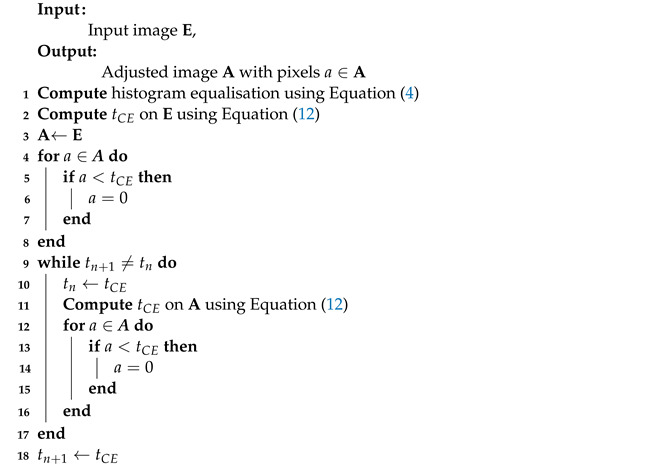


#### 3.1.1. Histogram Specification Using Histogram Equalisation

Histogram equalisation is the best-known application of histogram specification traditionally used as an image enhancement technique that yields a more balanced histogram and better contrast. When used on infrared images, it simulates the effect of reducing the dynamic range of the camera before a scene is captured. In this section, we are interested in creating the effect of reduced camera sensitivity to reduce the level of details in the image and enhance the pedestrians. The mathematical foundation of histogram equalisation is based on the idea that pixels in the original and equalised images can be regarded, respectively, as continuous random variables *X* and *Y* in the range of gray levels [0,L−1] and the normalised histogram as probability density function (PDF) [34]. It is a transform *T* of *X* into *Y*, which spreads gray levels over the entire scale, and each gray level is allotted an equal number of pixels. Therefore, *Y* is defined as follows:(1)$$Y=T\left(X\right)=\left(L-1\right)\int_{0}^{X}p_{X}\left(x\right)dx,$$
where $p_{X}$ is the PDF of the original image. *T*, therefore, is the cumulative distribution of *X* multiplied by (L−1).

The histogram $h_{f}$ of an image *f* with *L* gray levels in the range [0, 255] is given as follows:(2)$$h_{f}\left(l\right)=n_{l}$$
where $n_{l}$ is the number of pixels with *l* gray level. If the image has *m* pixels in total, the normalised histogram $p_{f}$ is calculated as follows:(3)$$p_{f}\left(l\right)=\frac{h_{f}\left(l\right)}{m}$$
where l=0,1,…,L−1. Given gray level *l*, this is equivalent to dividing each gray level $n_{l}$ by the total number of pixels in the image *m*. Given intensity *k*, the histogram equalised image *g* of *f* can then be defined by as follows.
(4)$$g\left(k\right)=\mathrm{floor}\left(L-1\right)\sum_{l=0}^{k}p_{f}\left(l\right)$$

#### 3.1.2. Histogram Partitioning

Histogram partitioning is normally carried out by finding a suitable gray value with which it is used to divide the histogram to identify and/or isolate objects of interest. Selecting the suitable gray value can be performed by using a local or global thresholding technique. Local techniques partition an image into subimages and determine a threshold for each of these subimages, while global techniques partition the entire image with a single threshold value. In this section, we are interested in finding the range at which the pedestrians lie, and this is performed by using iterative global thresholding, beginning with the histogram equalised image until convergence. At each stage of the iteration, the threshold value obtained becomes the new minimum dynamic range. The threshold value at each iteration is obtained using minimum cross-entropy. The details are given as follows.

The thresholding problem is the choice of the best distribution estimate for an event with unknown probabilities. Let the event with unknown probabilities $b^{+}\left(x_{i}\right)$ be the resulting binary image where $x_{i}$ refers to the grey level of the image pixels or the number of bins in the image histogram. The problem is to choose a distribution *b* that best estimates $b^{+}$ given what we know. The solution to the problem is the distribution having expected values that fall within the bounds or equal to the known values, thereby satisfying certain learned expectations $\sum_{i}b^{+}\left(x_{i}\right)m_{d}\left(x_{i}\right)$ or constraints. However, there is an infinite set of distributions that satisfy the constraints. Information is a measure of one’s freedom of choice in making selections [35]; thus, entropy, a measure of information, becomes necessary. The principle of maximum entropy, which states that the distribution of choice, from all that satisfy the constraints, is the one with the largest entropy, is the prescribed solution for solving such problems. However, in situations where a prior distribution that estimates $b^{+}$ is known in addition to learned expectations, the principle of minimum cross-entropy, a generalisation of the maximum entropy principle, applies [36].

Let the original image be the prior distribution with known probabilities $w(x_{i})$. Cross-entropy $H_{CE}\left(b,w\right)$ is the average number of bits needed to encode data with distribution *b* when modelled with distribution *w* [37] and is defined as follows:$$H_{CE}\left(b,w\right)\;=\sum_{i}b\left(x_{i}\right)\log\frac{1}{w(x_{i})}\;=-\sum_{i}b\left(x_{i}\right)\log w(x_{i})$$
and can be written as
(5)$$H_{CE}\left(b,w\right)=H_{E}\left(b\right)+D_{KL}\left(b||w\right)$$
where the following:$$H_{E}\left(b\right)=-\sum_{i}b\left(x_{i}\right)\log b\left(x_{i}\right)$$
is the entropy of the distribution *b* and the following:$$D_{KL}\left(b||w\right)=\sum_{i}b\left(x_{i}\right)\log\frac{b(x_{i})}{w(x_{i})}$$
is the Kullback–Leibler (K-L) divergence of distributions *b* and *w* defined as the excess code over the optimal code needed to represent data because it was modelled using distribution *w* instead of true distribution *b* [37]. The principle of minimum cross-entropy states that of all distributions *b* that satisfy constraints, the distribution of choice is the one with the smallest cross-entropy [36]. The value of $H_{E}\left(b\right)$ in Equation (Equation 5) is fixed and during minimization and it reduces to an additive constant. Therefore, minimizing cross-entropy reduces to minimizing K-L divergence:(6)$$H_{CE}\left(b,w\right)=\sum_{i}b\left(x_{i}\right)\log\frac{b(x_{i})}{w(x_{i})}$$
which is subject to the following.
$$\sum_{i}b\left(w_{i}\right)=\sum_{i}w\left(x_{i}\right)\;\left(=1\right)$$

The Monkey Model proposed in [38] views a digital image as a discrete probability distribution. In the model, a troop of monkeys is responsible for randomly throwing balls onto an empty array of cells. The image formed shows the number of balls received at each cell in the array. This view is exactly how infrared images are formed as the pixel intensity is a measure of the level of infrared emission received from the scene by the thermal camera’s sensors. Normalizing pixel intensity provides an approximate measure of the probability of the emission at each pixel from the scene. Therefore, known probabilities $w(x_{i})$ of the original image are the normalised pixel intensity values.

The binary image’s distribution *b* is determined from that of original image *w* and can be described using two probability measures $\mu_{0}\left(T\right)$ and $\mu_{1}\left(T\right)$, which are the below—and above—threshold means of the original image, respectively. These expectations are summarised as follows:(7)$$\begin{array}{c} \begin{array}{cc} (i)\;\; & b\left(x_{i}\right)\in \left\{\mu_{0}\left(T\right),\mu_{1}\left(T\right)\right\}\\ (ii)\;\; & \sum_{w(x_{i})<T}w\left(x_{i}\right)=\sum_{i<T}\mu_{0}\left(T\right)\\ (iii)\;\; & \sum_{w(x_{i})\geq T}w\left(x_{i}\right)=\sum_{i\geq T}\mu_{1}\left(T\right) \end{array} \end{array}$$
which allows for the determination of $\mu_{0}\left(T\right)$ and $\mu_{1}\left(T\right)$ as follows:(8)$$\begin{array}{c} \begin{array}{cc} \mu_{0}\left(T\right) & =\sum_{i=y}^{T}\;ip_{i}\\ \mu_{1}\left(T\right) & =\sum_{i=T+1}^{z}\;ip_{i} \end{array} \end{array}$$
where *y* and *z* are the smallest and largest grey levels present in the original image, respectively, *T* is the candidate threshold value, and $p_{i}$ is the probability of grey level *i* given by the following:(9)$$P_{i}=\frac{m_{i}}{N}$$
where $m_{i}$ is the number of pixels having grey level $x_{i}$, and *N* is the total number of pixels making up the image. Therefore, we have the following.
(10)$$\sum_{i=y}^{i=z}w\left(x_{i}\right)=\sum_{y}^{i<T}\mu_{0}\left(T\right)+\sum_{i\geq T}^{z}\mu_{1}\left(T\right)$$

Hence, cross entropy from Equation (Equation 6) becomes the following:(11)$$\begin{array}{c} \begin{array}{cc} H_{CE}\left(b,w\right)=\sum_{i=y}^{T}\mu_{0}\left(T\right) & \log\left(\frac{\mu_{0}\left(T\right)}{w(x_{i})}\right)\\ & +\sum_{i\geq T}^{z}\mu_{1}\left(T\right)\log\left(\frac{\mu_{1}\left(T\right)}{w(x_{i})}\right) \end{array} \end{array}$$
and the threshold to be chosen corresponds to the minimum cross entropy and is provided as follows.
(12)$$t_{CE}=arg\left(\min_{y<T\leq z}\left\{H_{CE}\left(b,w\right)\right\}\right)$$

Since we know that humans exist in the brightest part of the image, we obtain the iterative procedure shown in Algorithm 1 until iteration converges. The convergence condition is as follows:(13)$$t_{n+1}=t_{n}$$
where $t_{n+1}$ is the current $t_{CE},$ while $t_{n}$ is the previous one.

### 3.2. Candidate Validation

With background effectively suppressed, the resulting image is binarised, and the white regions are compared with the original image to separate from the remaining background blobs. This is performed by using standard deviation because humans do not have uniformly distributed intensities. This means that they have a higher standard deviation than noise and other sources of high infrared emission that also appear as hot spots. The threshold value for standard deviation is determined empirically for each video because the videos were acquired under different weather conditions. Candidate regions with standard deviation less than the determined threshold are excluded from the results.

## 4. Experimental Results and Discussion

### 4.1. Dataset

The performance of the proposed method is tested by the following publicly available databases:OTCBVS benchmark—Ohio State University (OSU) thermal pedestrian database [39], which contains ten sessions of 360 × 240 thermal images of the walking intersection and street of the Ohio State University captured during both day and night over many days under a variety of environmental conditions culminating in a total of 284 frames, each having an average of three to four people. The images were captured using Raytheon 300D thermal sensor with a 75 mm lens camera mounted on an eight-storey building.LITIV dataset [40], which contains nine sequences of 320 × 240 thermal videos captured at 30 frames per second with different zoom settings from relatively high altitudes and at different positions culminating in a total of 6325 frames of lengths varying between 11 s and 88 s.OTCBVS benchmark—Terravic Motion IR database [41], which features 18 thermal sequences with 8-bit grayscale JPEG images of size 320 × 240 pixels taken with a Raytheon L-3 Thermal Eye. Eleven sequences were chosen from the Outdoor Motion and Tracking (OMT) Scenarios.The Linkoping Thermal InfraRed (LTIR) dataset [42], which consists of 20 thermal infrared sequences featured in the Visual Object Recognition (VOT) challenge 2015. Four sequences pertaining to pedestrian detection were chosen: Saturated, Street, Crossing, and Hiding.

### 4.2. Qualitative Performance Evaluation

In this section, the performance of the proposed candidate generation method is showcased under visible differences in image appearance. Generally, hotspot algorithms require a very wide contrast between the object and the background. However, we observed that the proposed algorithm produces very good results with different ranges of contrast between the pedestrian and the background.

Firstly, the difference between using minimum cross-entropy for threshold selection [31] and the proposed candidate generation algorithm is shown in Figure 4, Figure 5, Figure 6 and Figure 7. The images in the LITIV database (Figure 4) appear to have a uniformly dark background, but thresholding reveals this assumption to be false. Therefore, only pedestrians that are located within the centre of the image can be extracted for validation, while pedestrians at the corners are merged with false positive pixels detected as the foreground. However, the proposed method can extract pedestrians at any location in the image frame. The images in the LTIR database (Figure 5) have varying histograms widths. Again, we have an image that looks dark and uniform, but it surprisingly has a lot of details constrained within a few gray levels in the histogram (Figure 5, row 3 from the Crossing Sequence). Most of the other images in selected sequences pertaining to pedestrian detection have backgrounds with intensities competing with pedestrians and minimum cross-entropy is unable to separate the background from the pedestrians. The brightness of the background is also a problem for most of the sequences in the Terravic dataset (Figure 6). Thresholding with minimum cross entropy shows the most varied response when applied on the OSU Thermal database (Figure 7). While there appears to be sufficient contrast in row 2 of Figure 7, the thresholding algorithm is not able to partition the image appropriately. Instead, the entire image is classified as ROI. In all these scenarios, the proposed candidate generation method has been able to detect all likely regions where pedestrians are located. It has also been removing as much of the background as possible so that pedestrians can easily be validated.

Secondly, the detection results of the proposed method are shown in Figure 8, Figure 9 and Figure 10. It can be observed that the method produces good results, especially in the area of a reduction in false positives. In cases where the pedestrians are close and overlapping, the proposed method tends to use one bounding box to detect them, similarly to LTIR sequences (Figure 9). In Figure 10, it can be observed that, under light rain conditions, there is a general difficulty in detecting the feet of some pedestrians, which is likely because they have been cooled by rain. Under partly cloudy conditions, although the pedestrians are very bright, the challenge is that there are several objects equally as bright as the pedestrian. The worst performance is under hazy weather in the afternoon, and we can observe fragmentation in one of the detected pedestrians while only the upper body is detected in another.

### 4.3. Quantitative Performance Evaluation

Quantitative evaluation is performed using four measures—True Positive, False Positive, Precision, and Recall. True Positive (TP) measures the number of detections that are pedestrians. False Positive (FP) measures the number of detections that are not pedestrians. Recall computes the ratio of the total number of pedestrians detected correctly to the total number of pedestrians.
$$\mathrm{Recall}=\frac{\mathrm{TP}}{\mathrm{TP}\;+\;\mathrm{FN}}$$

Precision computes the ratio of true detections to the total number of detections.
$$\mathrm{Precision}=\frac{\mathrm{TP}}{\mathrm{TP}\;+\;\mathrm{FP}}$$

The OSU thermal dataset was first evaluated as it provides information about the weather conditions during the time of capture. Table 1 provides the weather conditions under which the video sessions were acquired such as cloud conditions, time of day (TOD), the amount of ultraviolet rays from the sun (UV), and the temperature in degrees Celsius. Table 2 shows the detection results of the proposed method on each of the sessions used for the experiment. As the proposed method relies on the premise that pedestrians are brighter than the background, the third session of the dataset was discarded from the experiments because the pedestrians were polar-reversed. Therefore, the total number of pedestrians used for the experiment is 883. The proposed method detected 867 pedestrians (TP) and 3 non-pedestrians (FP), thus achieving an average precision of 0.9967 and an average recall of 0.9814.

From the weather conditions in Table 1 and results in Table 2, the following can be observed about the performance of the proposed method. First, the proposed method performs best in the morning under partly cloudy and fair conditions, while the worst performance is observed in the afternoon under hazy conditions. Although both sessions 9 and 10 have hazy weather, session 9 only had a single visible pedestrian walking across the scene for most of the video while session 10 had between three and six pedestrians at any given time with varying levels of brightness. Therefore, we are inclined to proceed with the lower performance of the method under hazy conditions even though session 9 had excellent precision and recall. Second, the proposed method performs best under temperatures lower than 20 °C regardless of the time of day or cloud cover. This is because, although the clothing worn by pedestrians normally impedes infrared emissions detected from the body, causing the pedestrians to appear non-homogeneous, this is less of a problem at lower temperatures as the amount of infrared that escapes the clothing still appears brighter than the surrounding atmosphere. From the above observations, we conclude that while the proposed method performs an excellent job in detecting pedestrians under different weather conditions, the best time to use infrared for monitoring is when atmospheric temperatures are low.

In Table 3, we compare the performance of the proposed method using TP and FP with other methods in the literature tested on the OSU thermal dataset. Davis and Keck [39] and Oluyide et al. [43] employed background modelling for candidate generation while Manda et al. [19] employed histogram partitioning. Zhao et al. [21] utilised temperature maps for detection. From this table, it is interesting to observe that the results in these other methods follow the same trend as the proposed method. That is to say that the best results are in the morning and under cloud cover conditions of partly cloudy and fair weather and also at low temperatures below and/or around 20 °C. Therefore, it is safe to conclude that while major efforts have been made to accurately detect pedestrians, the prevailing weather conditions still affect the results. Moreover, given that the proposed method cumulative outperforms the methods we have put forward from the literature and has the highest number of pedestrians (or detected all the pedestrians) on seven out of the nine sessions, we conclude that our method provides excellent results in detecting pedestrians under different weather conditions.

Furthermore, we compare the precision and recall of the proposed method with methods using the state-of-the-art object detector trained and/or tested with images in the OSU thermal database and presented the results in Table 4. Huda et al. [24] trained the detector using a private infrared database while Krišto et al. [25] and Haider et al. [27] trained the detector using samples from the OSU database. It can be observed that the proposed method outperforms state-of-the-art YOLO-based human detection methods [24,25] and performs slightly better than [27]. Many state-of-the-art methods do not perform similarly on thermal infrared images for several reasons. Firstly, thermal images are not RGB-turned-grayscale images but tone-mapped radiometric data. Because the information measured and displayed by RGB and Infrared images are different, the same method applied to both usually will not yield the same level of performance. Secondly, most of the state-of-art used features learned from visible images to classify pedestrians in infrared images. However, the level of details and image quality are not the same, which also results in poorer performance.

Lastly, we present the quantitative evaluation of the proposed methods on sequences chosen from the LTIR database, the nine (9) sequences from the LITIV database, and the eleven (11) sequences from the Outdoor Motion and Tracking (OMT) scenarios in the Terravic Database using precision and recall in Table 5. While target detection is challenging due to the loss of radiometric information in the conversion to digital images, it is also challenging to generalise methods across IR images taken with different thermal sensors and under different conditions. In [25], a model trained using a private IR dataset was tested on public datasets including OSU thermal, and it was found that the dataset with the highest performance was the one closest to trained data. However, it can be observed that the proposed method does produce good results across different datasets, especially given that a lot of false regions have been removed by the candidate generation stage. The drawback to the proposed method is in cases of severe temperature differences caused by clothing and accessories blocking infrared emission from the pedestrian, leading to fragmented segmentations in the ROI extraction step.

## 5. Conclusions

In this study, a new approach for candidate generation or ROI extraction based on dynamic range adjustment was presented. The method makes use of a combination of histogram specification and partitioning, which provides a clear starting point, a way to move towards the brighter regions, and a clear convergence point guaranteed to effectively suppress the background so that candidate validation could be carried out. The results show the validity of our method to detect pedestrians with different visual appearances, thermal sensors, and different weather conditions. Future work will involve optimising the algorithm to improve processing times.

## Figures and Tables

**Figure 1 sensors-22-01728-f001:**
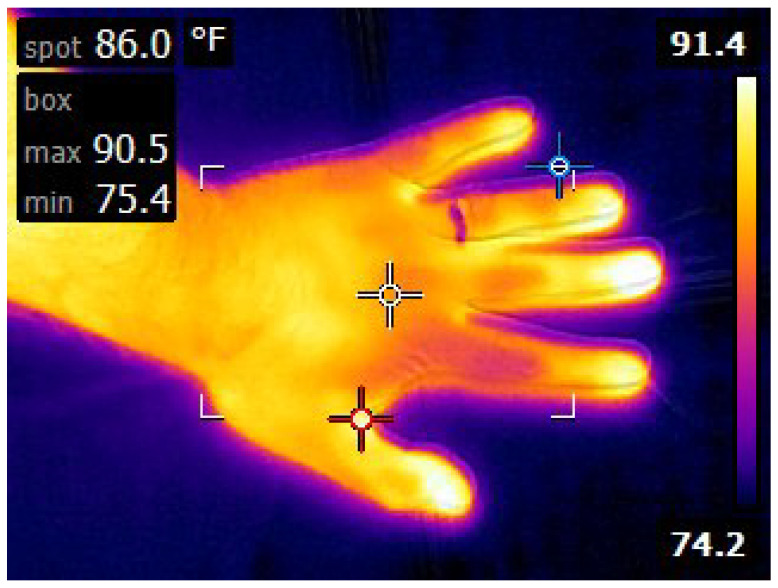
Example of how emissivity and reflected background can affect an infrared image. The ring appears cooler than the hand although they are both at the same temperature [3].

**Figure 2 sensors-22-01728-f002:**
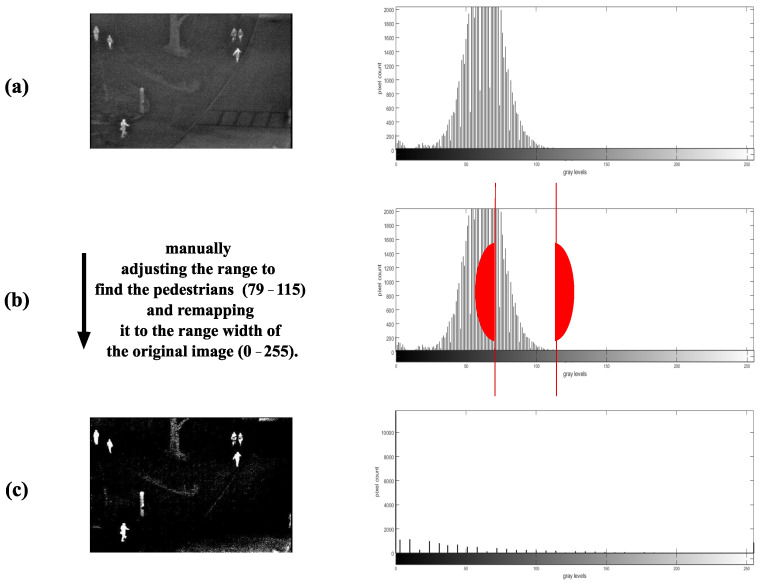
An example of manually adjusting the image to find the range of humans. (**a**) shows the original image and its histogram, and (**b**) shows the manually chosen range, which is between the two red lines (best viewed in colour). (**c**) shows the resulting image and the readjusted histogram.

**Figure 3 sensors-22-01728-f003:**
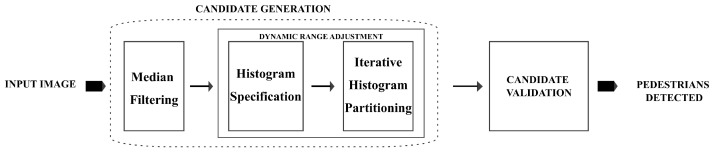
Overview of the proposed method.

**Figure 4 sensors-22-01728-f004:**
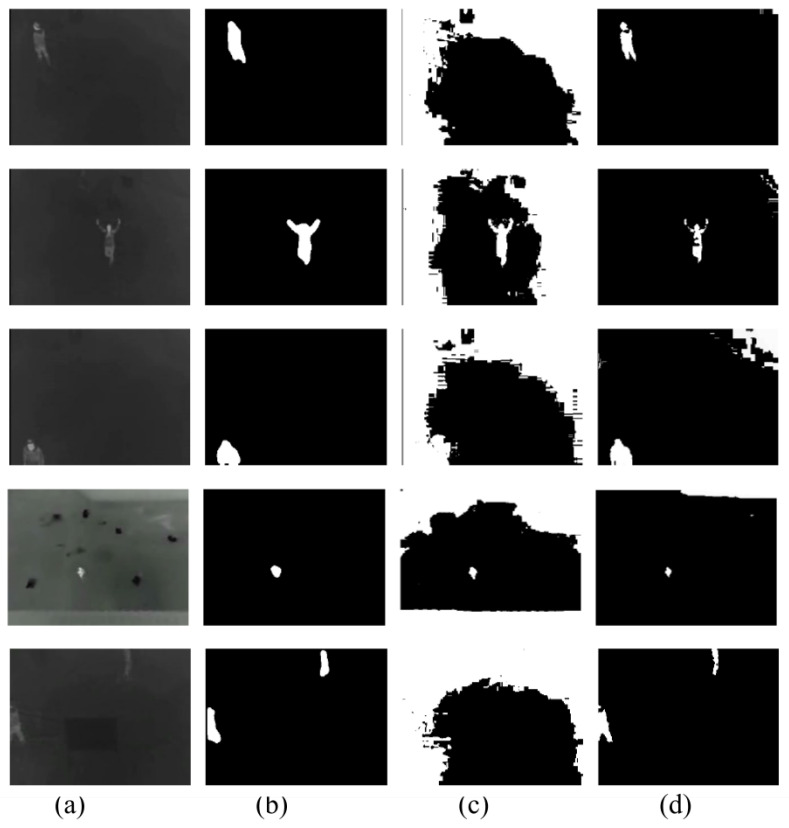
Comparison of outputs using MCET and the proposed method on images from the LITIV database. (**a**) Original image; (**b**) ground truth; (**c**) minimum cross entropy; (**d**) proposed method.

**Figure 5 sensors-22-01728-f005:**
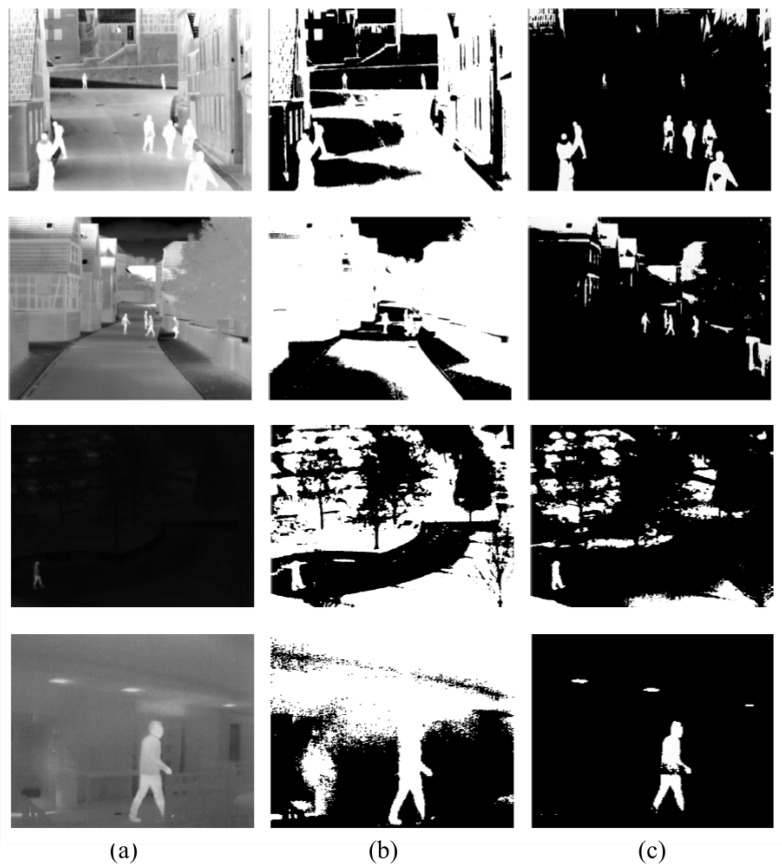
Comparison of outputs using MCET and the proposed method on images from the LTIR database. (**a**) Original image; (**b**) minimum cross entropy; (**c**) proposed method: Row 1: image 00000218 in Saturated; Row 2: image 00000005 in Street; Row 3: image 00000008 in Crossing; Row 4: image 00000136 in Hiding.

**Figure 6 sensors-22-01728-f006:**
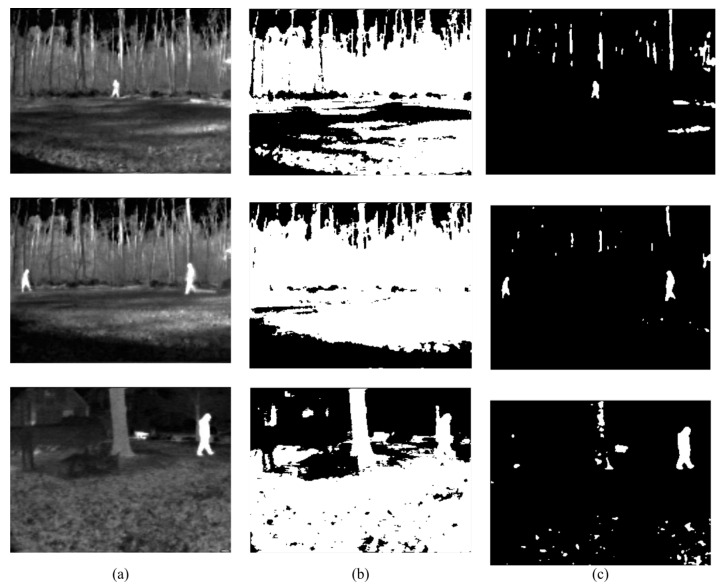
Comparison of outputs using MCET and the proposed method on images from the Terravic Thermal database. (**a**) Original image; (**b**) minimum cross entropy; (**c**) proposed method.

**Figure 7 sensors-22-01728-f007:**
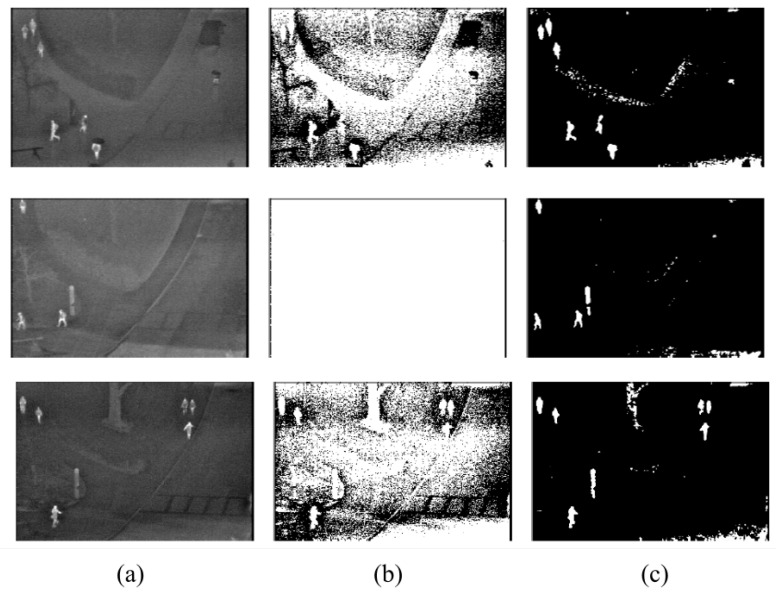
Comparison of outputs using MCET and the proposed method on images from the OSU Thermal database. (**a**) Original image; (**b**) minimum cross entropy; (**c**) proposed method.

**Figure 8 sensors-22-01728-f008:**
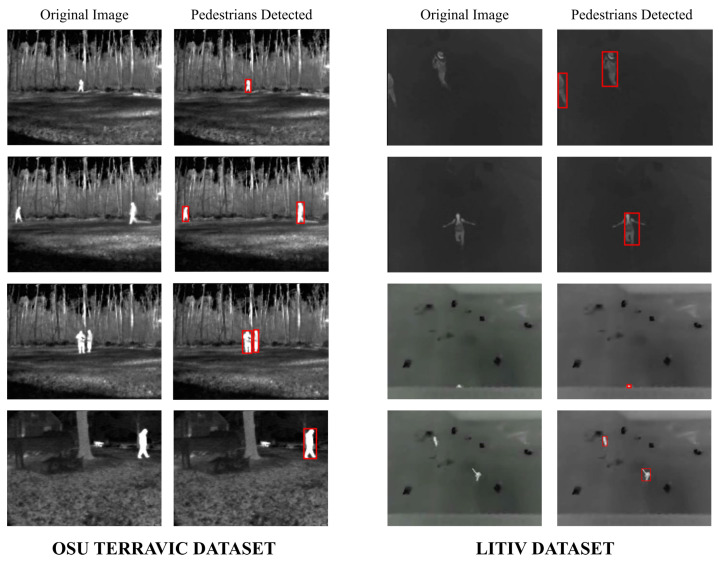
Detected pedestrians on images from the LITIV and Terravic dataset after validation.

**Figure 9 sensors-22-01728-f009:**
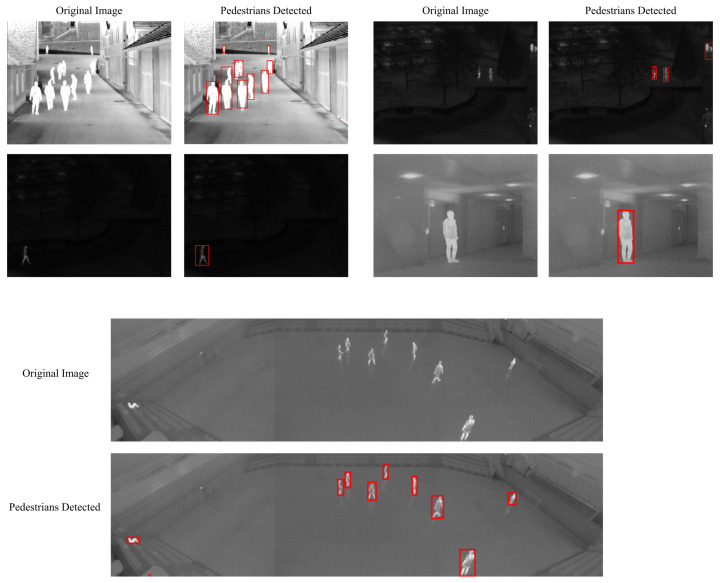
Detected pedestrians on images from the LTIR dataset after validation.

**Figure 10 sensors-22-01728-f010:**
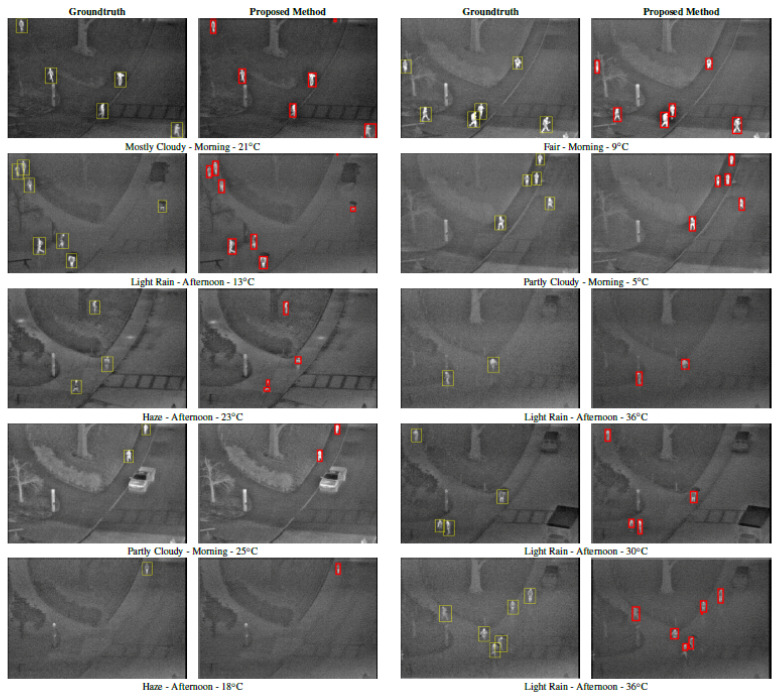
Detected pedestrians on images from the OSU thermal dataset after validation.

**Table 1 sensors-22-01728-t001:** Weather conditions for OSU thermal database.

Session	Cloud Condition	TOD	UV	Temp. (°C)
1	Light Rain	Afternoon	1	13
2	Partly Cloudy	Morning	1	5
4	Fair	Morning	4	9
5	Partly Cloudy	Morning	1	25
6	Mostly Cloudy	Morning	1	21
7	Light Rain	Afternoon	1	36
8	Light Rain	Afternoon	2	30
9	Haze	Afternoon	0	18
10	Haze	Afternoon	2	23

**Table 2 sensors-22-01728-t002:** Detection results of the proposed method.

Session	#People	#TP	#FP	Precision	Recall
1	91	88	0	1.0000	0.9670
2	100	100	0	1.0000	1.0000
4	109	109	0	1.0000	1.0000
5	101	101	2	0.9805	1.0000
6	97	94	0	1.0000	0.9690
7	80	93	1	0.9893	0.9893
8	96	98	0	1.0000	0.9898
9	95	95	0	1.0000	1.0000
10	97	89	0	1.0000	0.9175

**Table 3 sensors-22-01728-t003:** Comparison of TP and FP values of the proposed method with other methods.

		#TP					#FP				
	#Ped	[39]	[21]	[19]	[43]	Ours	[39]	[21]	[19]	[43]	Ours
1	91	88	77	78	85	88	0	3	0	0	0
2	100	94	99	98	97	100	0	2	2	2	0
4	109	107	107	109	109	109	1	7	10	0	0
5	101	90	97	101	97	101	0	16	16	1	2
6	97	93	92	97	93	94	0	8	0	0	0
7	94	92	78	80	90	93	0	8	0	1	1
8	99	75	89	96	93	98	1	8	0	0	0
9	95	95	91	95	95	95	0	4	16	0	0
10	97	95	91	83	89	89	3	18	6	0	0
1–10	883	829	821	837	848	867	5	74	50	4	3

**Table 4 sensors-22-01728-t004:** Comparison of precision and recall of the proposed method with the state-of-the-art.

Method	Precision	Recall
[25]	0.8600	0.8900
[24]	0.7100	0.6100
[27]	0.9920	0.9775
Ours	0.9967	0.9814

**Table 5 sensors-22-01728-t005:** Detection results on all datasets used in the study.

Database	Sequence	Precision	Recall
LTIR	Saturated	0.9845	0.9644
LTIR	Street	0.9590	0.9878
LTIR	Crossing	0.9990	0.9995
LTIR	Hiding	0.9833	0.9963
LITIV	All	0.9793	0.9891
Terravic	11 (OMT)	0.9818	0.9930
OSU	All	0.9967	0.9814

## Data Availability

The datasets generated and/or analysed during the current study are available in: Ohio State University (OSU) thermal pedestrian database (http://vcipl-okstate.org/pbvs/bench/Data/01/download.html) [39]; LITIV dataset (https://www.polymtl.ca/litiv/en/codes-and-datasets) [40]; Terravic Motion IR database (http://vcipl-okstate.org/pbvs/bench/Data/05/download.html) [41]; The Linkoping Thermal InfraRed (LTIR) dataset (https://www.cvl.isy.liu.se/en/research/datasets/ltir/version1.0/) [42].

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
