# Peer review of "Automatic Dynamic Range Adjustment for Pedestrian Detection in Thermal (Infrared) Surveillance Videos"

_sensors, 2022, doi:10.3390/s22051728_

Round 1

Reviewer 1 Report

The work subject is of importance in many scientific aspects. For image processing researchers - moving object detection, it presents a good discussion about the current state of the art. It also introduces good suggestions in that field. However,

1. the authors have not covered some also significant references like "Soharab Hossain Shaikh, Khalid Saeed, Nabendu Chaki (2014): Moving Object Detection using Background Subtraction. Springer in the series Springer-Brief in Computer Science," where good results were obtained and worth comparing with the authors results,

2. the authors have introduced a method for binarazation on the basis of iterative histogram partitioning. A similar approach was discussed in the mentioned above book and other works of the same authors. The authors should at least show how their methodology surpasses that method from some or all points of view.

3. The language is very good, but still there are some sentences that need a better contruction - sytax type, like very long sentences (an example is the first sentence in the  abstract - it is about 4 lines). There also some typos like: "... same temperature, However, based on ....".

Nevertheless, and in general, the work is well presented and the contents are of interest to the researchers in the field of moving object detection.

Author Response

Find responses attached

Reviewer 2 Report

This paper proposes an infrared pedestrian detection method based on histogram specification and iterative histogram partitioning to automatically adjust the dynamic range for candidate generation. Experimental results on four databases demonstrate the effectiveness of the proposed algorithm when compared with state-of-the-arts. Some comments are listed as follows:

  1. The introduction and related work can be divided into two sections.
  2. For describing the exiting works, according to the usage of background modelling, two categories are given. The references [3-7] belong to the first category, but [7] is introduced after the second category. Please make the introduction clearer.
  3. Why different methods are used to compare in those tables and figures? For example, table 3 lists [21], [7] and [26], while table 4 lists [11], [10] and [13].
  4. It is recommended to compare with the method proposed in: Infrared Pedestrian Detection with Converted Temperature Map.
  5. It would be better to compare the time complexity since it is the future work pointed out by authors.

Author Response

Find responses attached 

Round 2

Reviewer 2 Report

The authors have addressed my comments.